

# A lightweight speech recognition method with target-swap knowledge distillation for Mandarin air traffic control communications

Jin Ren[1,2], Shunzhi Yang[1], Yihua Shi[3] and Jinfeng Yang[1]

[1] Institute of Applied Artificial Intelligence of the Guangdong-Hong Kong-Macao Greater Bay Area, Shenzhen Polytechnic University, Shenzhen, Guangdong, China
[2] Shenzhen Institutes of Advanced Technology, Chinese Academy of Sciences, Shenzhen, Guangdong, China
[3] Industrial Training Centre, Shenzhen Polytechnic University, Shenzhen, Guangdong, China

## ABSTRACT

Miscommunications between air traffic controllers (ATCOs) and pilots in air traffic control (ATC) may lead to catastrophic aviation accidents. Thanks to advances in speech and language processing, automatic speech recognition (ASR) is an appealing approach to prevent misunderstandings. To allow ATCOs and pilots sufficient time to respond instantly and effectively, the ASR systems for ATC must have both superior recognition performance and low transcription latency. However, most existing ASR works for ATC are primarily concerned with recognition performance while paying little attention to recognition speed, which motivates the research in this article. To address this issue, this article introduces knowledge distillation into the ASR for Mandarin ATC communications to enhance the generalization performance of the light model. Specifically, we propose a simple yet effective lightweight strategy, named Target-Swap Knowledge Distillation (TSKD), which swaps the logit output of the teacher and student models for the target class. It can mitigate the potential overconfidence of the teacher model regarding the target class and enable the student model to concentrate on the distillation of knowledge from non-target classes. Extensive experiments are conducted to demonstrate the effectiveness of the proposed TSKD in homogeneous and heterogeneous architectures. The experimental results reveal that the generated lightweight ASR model achieves a balance between recognition accuracy and transcription latency.

## INTRODUCTION

To preserve aircraft safety and efficiency, air traffic controllers (ATCOs) and airline pilots must comprehend each other's intentions through radiotelephony communications with clarity and accuracy (*Lin, 2021*). In order for instructions to be transmitted and implemented correctly, it is crucial for both sides to make prompt, precise, and objective decisions about content integrity, terminology uniformity, and readback consistency.

Corresponding author
Yihua Shi, yhshi@szpt.edu.cn

Almost 83% of human information is perceived visually, roughly 11% auditorily, and the remainder is less than 6% (*Rosenblum, 2011*). Due to the fact that both ATCOs and airline pilots work in enclosed spaces with a limited field of vision, the effective visual perception signals consist primarily of displayed digits, radar scanning points, and other flight status indicators on the instrument panel of the air-ground data link system. Consequently, in contrast to the majority of situations, hearing now serves as the dominant means of immediate information perception during radiotelephony interactions (*Shi et al., 2022*). As control density has climbed, ATCOs suffer from heavier physical and mental burdens, particularly in light of the ongoing trend toward the explosive development of the air transportation business and the unprecedented growth of air traffic flow (*Guimin et al., 2018*). Occasionally, misunderstandings might occur due to noise interference, tiredness, distraction, or pressure (*Kim, Yu & Hyun, 2022*). Nevertheless, the consequences would be catastrophic in the case of an accident (*Helmke et al., 2021*).

Thanks to the tremendous advancement in fields of speech and language processing over the last decade, it is progressively becoming feasible for computing devices to automatically semantically verify ATCO's control commands and pilot's readback instructions. Ahead of back-end text matching for semantic verification, the audio signal is required to be transcribed into text *via* automatic speech recognition (ASR), which benefits the performance of the overall system. In addition, ASR can also alleviate the workload of air traffic controllers (ATCOs) (*Helmke et al., 2016*; *Ohneiser et al., 2021a*; *Ohneiser et al., 2021b*) and reduce the potential flight safety risks associated with misunderstandings.

To guarantee that ATCOs and pilots have sufficient time to respond instantly and efficiently, the ASR model is required to have both excellent recognition performance and low transcription latency. However, the majority of existing ASR works for air traffic control (ATC) (*Ohneiser et al., 2021a*; *Helmke et al., 2021*; *Zuluaga-Gomez et al., 2021*; *Lin et al., 2021c*) is mainly concerned with recognition performance while paying little attention to recognition speed, which motivates the research in this article. Over the past few years, the performance of end-to-end speech recognition models has been continuously enhanced owing to the expansion of model capacity and computational complexity (*Li & Etal, 2022*), leading to substantial computational expenses as well as time costs (*Georgescu et al., 2021*). In preparation for actual deployment on mobile or embedded devices, it is essential to construct a lightweight model that decreases the network size at the expense of negligible performance loss. Fortunately, advancements in knowledge distillation (*Li et al., 2021*; *Huang et al., 2018*) have shed light on this issue.

As a teacher-student training strategy, knowledge distillation (KD) aims to transfer knowledge from the heavy pre-trained teacher model into the relatively compact student model, regardless of their structural distinctions (*Wang & Yoon, 2021*). It can enhance the lightweight one's effectiveness and generalization without consuming extra expenses. Recently, decoupled knowledge distillation (DKD) (*Zhao et al., 2022*) has brought the logits-based KD approach back into the state-of-the-art, which can extract the rich "dark knowledge" (*Hinton, Vinyals & Dean, 2015*) from the teacher network efficiently by decoupling the standard KD loss into two separate parts, including target class knowledge distillation (TCKD) and non-target class knowledge distillation (NCKD). Note that the

term "target class" in KD refers to the specific class to which a given input sample belongs, *i.e.,* its ground truth label for training. DKD empirically explores and reveals that TCKD transfers knowledge regarding the difficulty of training examples, while NCKD is the key contributor to the success of logit distillation. In this manner, the original logits-based method's restrictions on the flexibility and effectiveness of knowledge transfer have been somewhat alleviated. Meanwhile, NCKD can play a more prominent role, especially when the target class encounters overconfidence. However, DKD still suffers from two issues: indirect strategies to mitigate overconfidence regarding the target class and insufficient investigation into hyperparameters for ASR tasks.

To alleviate the above issues, this article proposes a simple yet effective knowledge distillation strategy by swapping the logit output of the teacher and student models on the target class, named the Target-Swap Knowledge Distillation (TSKD). Precisely, it consists of two parts: the teacher's target class logit-swapped knowledge distillation (TKD) and the student's target class logit-swapped knowledge distillation (SKD). On the one hand, it can facilitate the student model to focus on NCKD; on the other hand, it can mitigate the potential overconfidence of the teacher model regarding the target class. In addition, hyperparameters between TKD and SKD are elaborately investigated through parameter sensitivity experiments for ASR tasks. Overall, our primary contributions are summarized as follows:

- To the best of our knowledge, we are pioneers in introducing the knowledge distillation idea into the ASR for Mandarin air traffic control communications. The generated lightweight ASR model balances recognition performance and transcription latency, allowing ATCOs and pilots sufficient time to respond immediately and effectively.
- This article proposes Target-Swap Knowledge Distillation (TSKD), a simple yet effective lightweight strategy that swaps the logit output of the teacher and student models for the target class. It can mitigate the potential overconfidence of the teacher model regarding the target class and enable the student model to concentrate on the distillation of knowledge from non-target classes.
- Extensive experiments are conducted to demonstrate the effectiveness of the proposed TSKD in several homogeneous and heterogeneous architectures.

The rest of the article is structured as follows. 'Related work' discusses the related work. 'Method' details the Target-Swap Knowledge Distillation strategy for Mandarin air traffic control communications. 'Experiments and result analysis' introduces the dataset and experimental implementation and analyzes the experimental results. 'Conclusions' concludes the research.

# RELATED WORK

## Automatic speech recognition for air traffic control communications
When it comes to automatic speech recognition for air traffic control communications, numerous works have been conducted by academia and industry from around the world. The majority of them concentrate on recognition performance, such as the recognition

error rate of words, characters, and sentences, which can reduce the workload of ATCOs and potential flight safety risks associated with miscommunications (*Helmke et al., 2016*).

Helmke and Ohneiser et al. adopted ontology-enhanced approaches for speech recognition applications in a variety of real-world ATC scenarios, such as robust command recognition and extraction (*Ohneiser et al., 2021a*; *Ohneiser et al., 2021b*) and callsign recognition for automatic readback error detection (*Helmke et al., 2021*). Zuluaga-Gomez et al. proposed a contextual semi-supervised learning approach combining untranscribed ATC speech and air surveillance data (*Zuluaga-Gomez et al., 2021*) and a two-stage method incorporating contextual radar data (*Nigmatulina et al., 2022*), which enhanced the callsign recognition. *Lin et al. (2021c)* proposed several end-to-end multilingual speech recognition frameworks for ATC communications (*Lin et al., 2021c*; *Yang et al., 2020*; *Lin et al., 2021a*; *Lin et al., 2021b*), which focus on the multilingual ASR on parallel Chinese and English ATC recordings. *Shi et al. (2022)* proposed an end-to-end Conformer-based multi-task learning speech recognition model for Mandarin radiotelephony communications in civil aviation, which can effectively extract global and local acoustic features, especially the contextual long-distance dependent local similarity features.

The above ASR works for air traffic control (ATC) is mainly concerned with recognition performance while paying little attention to recognition speed. To address this issue, this article introduces the lightweight idea of knowledge distillation into the ASR for Mandarin ATC communications to enhance the generalization performance of the light model. In this way, the generated lightweight ASR model achieves a balance between recognition performance and transcription latency, allowing ATCOs and pilots sufficient time to respond immediately and effectively.

## Knowledge distillation

Over the past decade, advancements in model compression (*Cheng et al., 2018*; *Choudhary et al., 2020*), such as knowledge distillation (*Li et al., 2021*; *Huang et al., 2018*), low-rank matrix factorization (*Povey et al., 2018*), network pruning (*Gao et al., 2020*), parameter quantization (*He et al., 2019*; *Sainath et al., 2020*), and lottery ticket hypothesis (*Ding, Chen & Wang, 2022*), have attracted much attention. Knowledge distillation (KD) aims to transfer knowledge from the heavy pre-trained teacher model into the relatively compact student model, regardless of their structural distinctions (*Wang & Yoon, 2021*). It can enhance the lightweight student model's generalization without introducing extra costs, since soft targets convey more generalization information than hard ones to prevent overfitting of the student model. Regarding knowledge source types, distilled knowledge consists of response-based knowledge from the final output layer, feature-based knowledge from intermediate layers, and relation-based knowledge within feature maps or data samples (*Gou et al., 2021*).

After the idea of KD was first proposed in *Hinton, Vinyals & Dean (2015)*, the logits-based KD and the feature-based KD, like FitNets (*Romero et al., 2015*), take turns playing the central role throughout the history of KD research. Compared with logits-based KD, feature-based KD achieves superior performance at the expense of extra computation and storage requirements during training. It is worth noting that decoupled

knowledge distillation (DKD) (*Zhao et al., 2022*) has recently brought the logits-based KD approach back into the state-of-the-art, which can efficiently extract the rich "dark knowledge" (*Hinton, Vinyals & Dean, 2015*) from the teacher network. However, DKD still suffers from two issues: indirect strategies to mitigate overconfidence regarding the target class and insufficient investigation into hyperparameters for ASR tasks. As for the former, with the help of an efficient decoupling mechanism, the conventional logits-based method's restrictions on the flexibility of knowledge transfer have been somewhat alleviated; therefore, NCKD can play a more prominent role, particularly when the target class encounters overconfidence. Nevertheless, it requires more direct strategies to prevent overconfidence in the target class (*Ren, Guo & Sutherland, 2022*). As for the latter, the scale factor hyperparameters for the two decoupled components differ considerably for various tasks. While DKD explored hyperparameters for a number of computer vision tasks, further investigation is required for assignments like ASR.

To address this issue, this article proposes Target-Swap Knowledge Distillation (TSKD), a simple yet effective lightweight strategy that swaps the logit output of the teacher and student models for the target class. On the one hand, it mitigates the potential overconfidence of the teacher model regarding the target class; on the other hand, it enables the student model to concentrate on the distillation of knowledge from non-target classes. In addition, hyperparameters between TKD and SKD are elaborately investigated through parameter sensitivity experiments for ASR tasks.

## METHOD

### The overall architecture of target-swap knowledge distillation (TSKD)

In logits-based offline knowledge distillation for speech recognition, the teacher model is a sophisticated pre-trained model with good performance. Their output logits are valuable prior knowledge, which can guide the student model to learn and represent the mapping relationship between acoustic features and predicted text through knowledge transfer.

In this article, we propose a simple yet effective lightweight strategy named Target-Swap Knowledge Distillation (TSKD), which swaps the logit output of the teacher and student models for the target class. Figure 1 depicts the overall architecture of knowledge distillation from the pre-trained teacher model to the student model with the TSKD strategy. The overall loss $\mathcal{L}_{overall}$ comprises two components, $\mathcal{L}_{TSKD}$ for the TSKD loss proposed in this article and $\mathcal{L}_{CE}$ for the original cross-entropy loss, which can be calculated as the formula below,

$$\mathcal{L}_{overall} = \alpha \mathcal{L}_{TSKD} + \beta \mathcal{L}_{CE}, \tag{1}$$

where $\alpha$ and $\beta$ are tuned hyper-parameters to make a trade-off between the two losses, given the usual constraints that $\alpha + \beta = 1$.

In the following two subsections, we will describe the two parts of TSKD: the teacher's target class logit-swapped knowledge distillation (TKD) and the student's target class logit-swapped knowledge distillation (SKD). On the one hand, it can enable the student model to concentrate on the distillation of knowledge from non-target classes; on the other
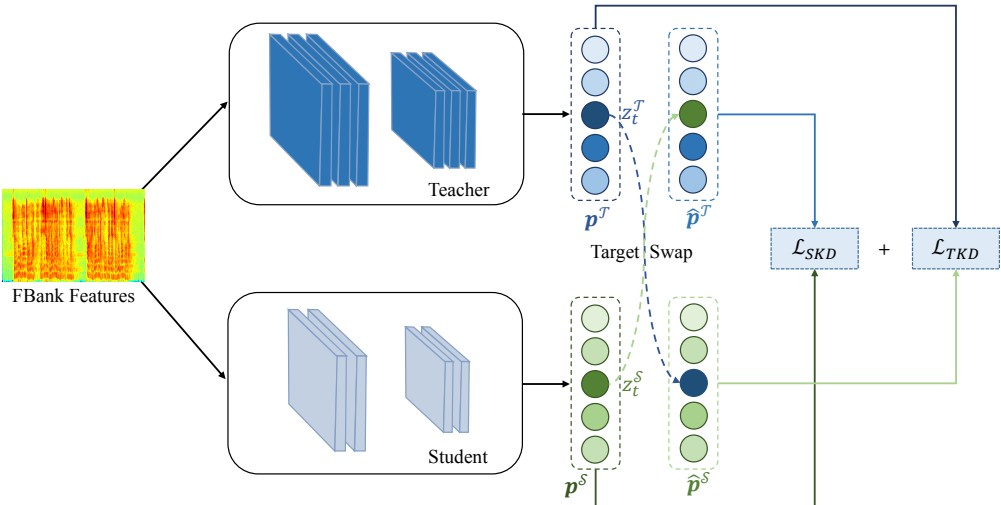

**Figure 1** The overall architecture of target-swap knowledge distillation (TSKD).

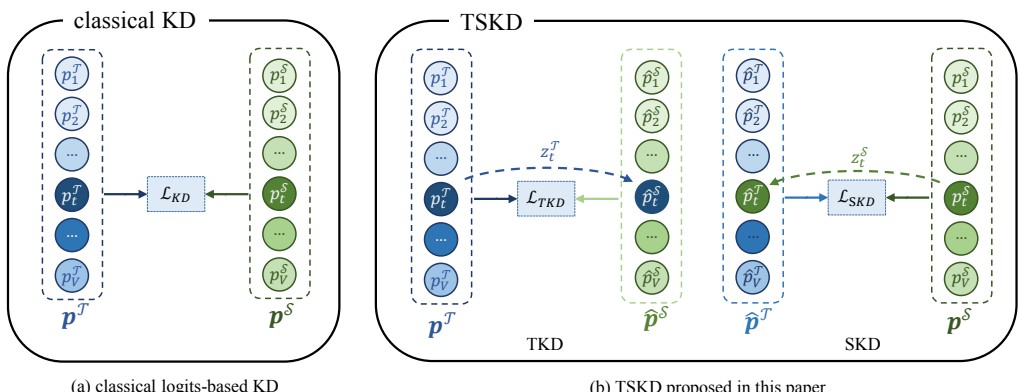

(a) classical logits-based KD      (b) TSKD proposed in this paper

**Figure 2** The distinction between the conventional logits-based knowledge distillation (*Hinton, Vinyals & Dean, 2015*) and TSKD proposed in this article.

hand, it can mitigate the potential overconfidence of the teacher model regarding the target class. Figure 2 illustrates the distinction between the conventional logits-based knowledge distillation loss function (*Hinton, Vinyals & Dean, 2015*) and that of the TSKD proposed in this article. In this way, the network will be optimized collaboratively to balance the two portions of enhanced loss functions, as calculated with the following formula:

$$\mathcal{L}_{TSKD} = \lambda_1 \mathcal{L}_{TKD} + \lambda_2 \mathcal{L}_{SKD}. \tag{2}$$

## Teacher's target class logit-swapped knowledge distillation (TKD)

With the acoustic features of phonetic fraction as input, the teacher model for speech recognition will generate the logits vector $\boldsymbol{z}^{\mathcal{T}} = \{z_1^{\mathcal{T}}, z_2^{\mathcal{T}}, \ldots, z_V^{\mathcal{T}}\}$, where $z_i$ represents the

**Peer**J Computer Science

[1] The temperature (T) in *Hinton, Vinyals & Dean (2015)* is omitted without loss of generality.

network's predicted logit value for the $i$-th character and $V$ represents the vocabulary size, *i.e.*, the total number of categories. Correspondingly, the student model will produce the logits vector $z^{\mathcal{S}} = \{z_1^{\mathcal{S}}, z_2^{\mathcal{S}}, ..., z_V^{\mathcal{S}}\}$, where $\mathcal{T}$ and $\mathcal{S}$ stand for the teacher and the student model, respectively. Thus, the class probability distributions of the teacher and student models, namely $\boldsymbol{p}^{\mathcal{T}} = \{p_1^{\mathcal{T}}, p_2^{\mathcal{T}}, ..., p_V^{\mathcal{T}}\}$ and $\boldsymbol{p}^{\mathcal{S}} = \{p_1^{\mathcal{S}}, p_2^{\mathcal{S}}, ..., p_V^{\mathcal{S}}\}$, can be calculated with the following formula [1] :

$$p_i = \frac{exp(z_i)}{\sum_{j=1}^{V} exp(z_j)}. \tag{3}$$

To urge the student model to concentrate on non-target classes for knowledge distillation, inspired by DKD (*Zhao et al., 2022*), the logit output of the student model on the target class is substituted with that of the teacher model $z_t^{\mathcal{T}}$. Hence, the student model yields the logits vector $z^{\mathcal{S}} = \{z_1^{\mathcal{S}}, z_2^{\mathcal{S}}, ..., z_{t-1}^{\mathcal{S}}, z_t^{\mathcal{T}}, z_{t+1}^{\mathcal{S}}, ..., z_V^{\mathcal{S}}\}$, and the student model's class probability can be represented as $\hat{\boldsymbol{p}}^{\mathcal{S}} = \{\hat{p}_1^{\mathcal{S}}, \hat{p}_2^{\mathcal{S}}, ..., \hat{p}_t^{\mathcal{S}}, ..., \hat{p}_V^{\mathcal{S}}\}$, which can be calculated as follows,

$$\hat{p}_t^{\mathcal{S}} = \frac{exp(z_t^{\mathcal{T}})}{\sum_{j=1, j \neq t}^{V} exp(z_j^{\mathcal{S}}) + exp(z_t^{\mathcal{T}})}, \tag{4}$$

$$\hat{p}_i^{\mathcal{S}} = \frac{exp(z_i^{\mathcal{S}})}{\sum_{j=1, j \neq t}^{V} exp(z_j^{\mathcal{S}}) + exp(z_t^{\mathcal{T}})}, \tag{5}$$

where $i \in \{1, 2, ..., t-1, t+1, ..., V\}$. Given that the logits generated by the teacher model remain unchanged, its class probability distribution also keeps identical, namely $\boldsymbol{p}^{\mathcal{T}} = \{p_1^{\mathcal{T}}, p_2^{\mathcal{T}}, ..., p_V^{\mathcal{T}}\}$. The student model's modified class probability distribution $\hat{\boldsymbol{p}}^{\mathcal{S}}$ and that of the teacher model $\boldsymbol{p}^{\mathcal{T}}$ are employed for the calculation of the KL divergence loss, named the Teacher's target class logit-swapped Knowledge Distillation (TKD), as the following formula:

$$\mathcal{L}_{TKD} = \boldsymbol{KL}(\boldsymbol{p}^{\mathcal{T}} || \hat{\boldsymbol{p}}^{\mathcal{S}})$$
$$= p_t^{\mathcal{T}} log\left(\frac{p_t^{\mathcal{T}}}{\hat{p}_t^{\mathcal{S}}}\right) + \sum_{i=1, i \neq t}^{V} p_i^{\mathcal{T}} log\left(\frac{p_i^{\mathcal{T}}}{\hat{p}_i^{\mathcal{S}}}\right). \tag{6}$$

### Student's target class logit-swapped knowledge distillation (SKD)

To alleviate the potential overconfidence (*Ren, Guo & Sutherland, 2022*) of the teacher model on the target class, the logit output of the teacher model on the target class is substituted with that of the student model $z_t^{\mathcal{S}}$. Thus, the teacher model produces the logits vector $z^{\mathcal{T}} = \{z_1^{\mathcal{T}}, z_2^{\mathcal{T}}, ..., z_{t-1}^{\mathcal{T}}, z_t^{\mathcal{S}}, z_{t+1}^{\mathcal{T}}, ..., z_V^{\mathcal{T}}\}$, and the teacher model's class probability distribution can be represented as $\hat{\boldsymbol{p}}^{\mathcal{T}} = \{\hat{p}_1^{\mathcal{T}}, \hat{p}_2^{\mathcal{T}}, ..., \hat{p}_t^{\mathcal{T}}, ..., \hat{p}_V^{\mathcal{T}}\}$, which can be calculated as follows,

$$\hat{p}_t^{\mathcal{T}} = \frac{exp(z_t^{\mathcal{S}})}{\sum_{j=1, j \neq t}^{V} exp(z_j^{\mathcal{T}}) + exp(z_t^{\mathcal{S}})}, \tag{7}$$

$$\hat{p}_i^{\mathcal{T}} = \frac{exp(z_i^{\mathcal{T}})}{\sum_{j=1,j\neq t}^{V} exp(z_j^{\mathcal{T}}) + exp(z_t^{\mathcal{S}})}, \tag{8}$$

where $i \in \{1, 2, \ldots, t-1, t+1, \ldots, V\}$. The teacher model's modified class probability distribution $\hat{p}^{\mathcal{T}}$ and that of the student model $p^{\mathcal{S}}$ are employed for the calculation of the KL divergence loss, named the Student's target class logit-swapped Knowledge Distillation (SKD), as the following formula:

$$\mathcal{L}_{SKD} = KL(\hat{p}^{\mathcal{T}} || p^{\mathcal{S}})$$
$$= \hat{p}_t^{\mathcal{T}} log\left(\frac{\hat{p}_t^{\mathcal{T}}}{p_t^{\mathcal{S}}}\right) + \sum_{i=1,i\neq t}^{V} \hat{p}_i^{\mathcal{T}} log\left(\frac{\hat{p}_i^{\mathcal{T}}}{p_i^{\mathcal{S}}}\right). \tag{9}$$

# EXPERIMENTS AND RESULT ANALYSIS

## Dataset

Under the instruction of first-line ATCOs and tutors of ATC courses, a corpus was established, and qualified ATCOs were chosen to record the corpus in a quiet situation. Consequently, the built dataset follows the linguistic and acoustic feature distribution of air-ground communications regarding call content, voice control, pronunciation regulations, syntactic structure, and linguistic conventions. The Mandarin ATC communications dataset is a paraphrase dataset, with each sample consisting of the ATCO's control instruction and the flight crew's readback instruction successively. In this double-checked manner, the command could be carried out efficiently and precisely.

The Mandarin ATC communications dataset is stored as a WAV audio file with a sampling frequency of 16 kHz, 16 bits, and mono for each recording. Ten males and seven females, a total of seventeen certified ATCOs, participated in the recording. With 641 corpora and 10971 recorded voices, the total speech data has a duration of 1516 minutes. The dataset is randomly scrambled and split into the training set, validation set, and test set in the ratio of 7:1:2.

## Implementation details

In this experiment, 80-dimensional FBank (Filter bank) features are taken as the model input, and frame window and frameshift size are 25ms and 10ms, respectively. The Kaldi toolkit (*Povey et al., 2011*) was employed to extract the acoustic features, and the speed factors of 0.9x and 1.1x were applied to scale the dataset for data augmentation. In the training procedure, the relative positional encoding of Transformer-XL (*Dai et al., 2019*) was utilized, and the Adam optimizer with the Noam learning rate (*Vaswani et al., 2017*) was adopted for efficient parameter optimization. All models were trained on the dataset through eight NVIDIA A100-SXM4-40G GPUs.

Concerning the effect of the teacher and student architectures on the efficacy of knowledge distillation, *Cho & Hariharan (2019)* reveals that an excellent teacher only sometimes produces qualified students due to their capacity gap. Hence, it is crucial to

**Table 1** Parameter configurations for the encoder and decoder of the student model variants.

| Base Model | Variants Abbreviations | Encoder | | | | Decoder | |
|---|---|---|---|---|---|---|---|
| | | N | d_model | d_ff | h | Attention Dimension | Hidden units Dimension |
| BiLSTM | BiLSTM_3_512 | 3 | 256 | – | – | 256 | 512 |
| Transformer | Trans_6_1024 | 6 | 256 | 2,048 | 4 | 512 | 1,024 |
| | Trans_12_512 | 12 | 128 | 1,024 | 2 | 256 | 512 |
| Conformer | Con_12_512 | 12 | 128 | 256 | 2 | 256 | 512 |
| | Con_12_256 | 12 | 128 | 256 | 2 | 256 | 256 |

select a suitable network as the student model given the teacher model. In this experiment, the teacher models adopted the bidirectional long short-term memory (BiLSTM) network, Transformer (*Vaswani et al., 2017*), and Conformer (*Gulati et al., 2020*) models, which followed the configurations in our previous work (*Shi et al., 2022*). To choose student models with appropriate capacity, several speech recognition model variants with varying network sizes were constructed by modifying the encoder and decoder structures based on BiLSTM, Transformer, and Conformer, respectively. For easy identification, individual abbreviations were assigned to the corresponding student models in Table 1. Each abbreviation consists of the base model type, the number of layers of the encoder $N$, and the dimension of the decoder's hidden units, which are connected by underscores.

The experiments employed characters as the fundamental modeling unit and the sentence error rate (SER) as the evaluation criterion with the following formula:

$$SER = \frac{F}{M}, \tag{10}$$

where $F$ is the number of sentences with transcription errors, and $M$ is the total number of sentences.

Table 2 represents the parameters, recognition error rate, and transcription time for all teacher and student model variants on the test set. For fairness, all the transcription results were calculated, given a beam search size of 10. Table 2 demonstrates that as model capacity decreases, the recognition performance of each of the three groups of models declines. Specifically, as the model capacity falls, the sentence error rate (SER) of the Transformer model rises sharply, whereas it goes up relatively gently for the LSTM and Conformer models. Notably, when the student model's capacity drops to a quarter of the teacher model's, the LSTM and Conformer models can still make reliable predictions. In contrast, the Transformer model can no longer make effective estimations. It verifies that the Conformer-based speech recognition model (*Shi et al., 2022*) can effectively extract global and local acoustic features, especially the long-distance context-dependent local similarity features in the dataset of Mandarin ATC communications. Overall, the Conformer-based knowledge distillation significantly lowers the model capacity with minimal accuracy loss.

**Table 2** Parameters, recognition error rate, and transcription time of the teacher and student model variants given a beam search size of 10.

|  | Model | Parameter (M) | SER (%) | Transcription time (ms) |
|---|---|---|---|---|
| Teacher | BiLSTM | 43.90 | 5.58 | 984 |
|  | Transformer | 34.28 | 4.12 | 1867 |
|  | Conformer | 36.71 | 2.89 | 2012 |
| Student | BiLSTM_3_512 | 12.39 | 7.32 | 816 |
|  | Trans_6_1024 | 26.39 | 11.57 | 1454 |
|  | Trans_12_512 | 9.74 | 11.85 | 934 |
|  | Con_12_512 | 9.91 | 4.35 | 1014 |
|  | Con_12_256 | 7.35 | 6.85 | 711 |

**Table 3** Effectiveness of TSKD in homogeneous and heterogeneous architectures.

| Method | SER (%) | |
|---|---|---|
| Teacher | Conformer | Transformer |
|  | 2.89 | 4.12 |
| Student | Con_12_256 | Con_12_512 |
|  | 6.85 | 4.35 |
| TKD | 6.39 | 4.07 |
| SKD | 6.67 | 4.35 |
| TSKD (Ours) | **6.00** | **3.98** |

**Notes.**
Results of the proposed model are in bold.

## Effectiveness of TSKD in homogeneous and heterogeneous architectures

Experiments were carried out on the Mandarin ATC communications dataset to investigate the effectiveness of the TSKD training strategy proposed in this article. As shown in Table 3, two distinct teacher-student network structures, including homogeneous and heterogeneous architectures, were employed. The left set used the Conformer teacher model to distill the Conformer student model within homogeneous architectures, whereas the right group made use of the Transformer teacher model to distill the Conformer student model within heterogeneous network structures. The findings suggest that the effectiveness of knowledge distillation can be boosted by employing either TKD or SKD, with TKD bringing greater benefits. It also supports that non-target class knowledge distillation (NCKD) is the primary reason why classical logits-based distillation is effective but severely restricted, which coincides with the efficiency analysis of DKD (*Zhao et al., 2022*). Furthermore, integrating two training strategies into one, namely TSKD, can further strengthen the performance of knowledge distillation.

In this article, TSKD focuses on the logit output exchange of the target class between the student and teacher models, which can effectively improve the distillation performance. Apart from the target class, the top-k information predicted by the teacher model for each segment of acoustic features is also a sort of valuable information (*Reddi et al., 2021*); it reflects which k characters the acoustic features input should be transcribed into, from the

**Table 4 Knowledge distillation performance through swapping top-k predictive information.**

| Method | SER (%) | |
|---|---|---|
| Teacher | Conformer | Transformer |
| | 2.89 | 4.12 |
| Student | Con_12_256 | Con_12_512 |
| | 6.85 | 4.35 |
| top-1 | 6.30 | 4.26 |
| top-2 | 6.48 | 4.38 |
| top-3 | 6.57 | 4.82 |
| top-5 | 6.20 | 4.44 |
| top-8 | 6.20 | 4.17 |
| top-10 | 6.57 | 4.17 |
| TSKD (Ours) | **6.00** | **3.98** |

**Notes.**
Results of the proposed model are in bold.

**Table 5 Performance comparison of various knowledge distillation methods in homogeneous architectures.**

| Method | SER (%) | | |
|---|---|---|---|
| Teacher | Conformer | Transformer | BiLSTM |
| | 2.89 | 4.12 | 5.58 |
| Student | Con_12_256 | Trans_12_512 | BiLSTM_3_512 |
| | 6.85 | 11.85 | 7.32 |
| classical KD (*Hinton, Vinyals & Dean, 2015*) | 6.30 | 11.76 | 6.94 |
| FitNets (*Romero et al., 2015*) | 6.22 | 10.92 | 6.68 |
| DKD (*Zhao et al., 2022*) | 6.11 | 11.39 | 6.30 |
| TSKD (Ours) | **6.00** | **10.65** | **6.11** |

**Notes.**
Results of the proposed model are in bold.

perspective of the teacher model. On the basis of this assumption, another comparative experiment was conducted to verify the effectiveness of the TSKD, with the predictive top-k output of the teacher model for each segment as the information exchange. As shown in Table 4, the experimental results show that exchanging top-k information is an effective knowledge distillation approach, nevertheless, its performance does not surpass that of the target class, *i.e.,* TSKD.

To further validate the effectiveness of TSKD proposed in this article, several typical knowledge distillation methods were implemented in homogeneous and heterogeneous network structures. The experimental results are presented in Tables 5 and 6, respectively. The results demonstrate that under these six homogeneous or heterogeneous architectures, the TSKD enhanced the recognition accuracy of all student models by an average of 1.13%, achieving the best distillation results in most experiments.

Specifically, the TSKD dramatically boosts the Transformer student model under heterogeneous architectures, reducing the SER by 2.13%. A plausible explanation is that the Conformer teacher model may enable the Transformer student model to capture

**Table 6** Performance comparison of various knowledge distillation methods in heterogeneous architectures.

| Method | SER (%) | | |
|---|---|---|---|
| Teacher | Conformer | Conformer | Transformer |
| | 2.89 | 2.89 | 4.12 |
| Student | Trans_6_1024 | BiLSTM_3_256 | Con_12_512 |
| | 11.57 | 7.32 | 4.35 |
| classical KD (*Hinton, Vinyals & Dean, 2015*) | 10.83 | 6.48 | 4.26 |
| FitNets (*Romero et al., 2015*) | 9.22 | 6.61 | 3.98 |
| DKD (*Zhao et al., 2022*) | **8.80** | 6.57 | 4.17 |
| TSKD (Ours) | 9.44 | **6.30** | **3.98** |

Notes.
The bold values indicate the best recognition performance in the corresponding column regarding SER (%).

**Table 7** Parameter sensitivity experiment between TKD ($\lambda_1$) and SKD ($\lambda_2$) in Equation 2.

| TKD ($\lambda_1$) | SKD ($\lambda_2$) | SER (%) |
|---|---|---|
| 1.8 | 0.2 | 4.07 |
| 1.5 | 0.5 | 4.07 |
| 1.2 | 0.8 | 4.44 |
| 1 | 1 | **3.98** |
| 0.8 | 1.2 | 4.07 |
| 0.5 | 1.5 | 5.00 |
| 0.2 | 1.8 | 4.73 |

Notes.
The bold values indicate the best recognition performance in the corresponding column regarding SER (%).

local similarity features more effectively, particularly the long-distance context-dependent local similarity features in the Mandarin ATC communications dataset. Notably, in the heterogeneous network employing the Transformer teacher model to distill the Conformer student model, the distillation performance of the student model exceeds that of the teacher; it suggests that Conformer is superior to Transformer in terms of the model architecture's efficient feature extraction capabilities.

## Parameter sensitivity experiment

To determine the relative importance between TKD and SKD, parameter sensitivity experiments were conducted to explore the influence of the proportion of the two components of TSKD on recognition accuracy. Table 7 presents the effect of varying ratios of the two parts of the loss functions in Eq. (2) on the SER of the student model, with Transformer and Con_12_512 serving as the teacher and student model, respectively. The experimental results show that the optimal knowledge distillation effect of 3.98% is achieved when the ratio of TKD to SKD is 1:1, indicating that the two parts of TKD and SKD are of equal value for effective knowledge distillation.

**Table 8** **Effect of beam search size on sentence error rate (SER (%)) and transcription time (ms).**

| Model | Beam search size | | | |
|---|---|---|---|---|
| | 1 | 3 | 5 | 10 |
| BiLSTM_3_512 | 6.21/396 | 6.19/546 | 6.13/784 | **6.11**/873 |
| Trans_6_1024 | 9.13/609 | 8.86/760 | 8.80/1213 | **8.80**/1531 |
| Trans_12_512 | 10.90/316 | 10.71/418 | 10.68/591 | **10.65**/984 |
| Con_12_512 | 4.29/365 | 4.11/465 | 3.99/628 | **3.98**/1034 |
| Con_12_256 | 6.23/324 | 6.07/409 | **6.00**/450 | 6.01/762 |

**Notes.**
The bold values indicate the best recognition performance in the corresponding column regarding SER (%).

### Effect of beam search size on recognition accuracy and transcription speed

The efficiency and accuracy of speech recognition depend on the beam search size as well as the model capacity. Table 8 details the effect of beam search size on recognition accuracy and transcription speed of the speech recognition student model variants. The results indicate that expanding the beam search size generally decreases SER, considering the models have a greater chance of predicting the optimal sequence of characters. However, this enhancement comes at the cost of a substantial rise in transcription time.

Specifically, when the beam search size is less than 3, the SER decreases dramatically as it increases. In contrast, when it is greater than 3, the SER decreases slowly and even increases in certain instances. When the beam search size is 10, transcription takes approximately twice as long as when it is 3. Nevertheless, the performance gain is limited to a maximum of 0.13% (Con_12_512) absolute recognition accuracy. In a word, by restricting the beam search size from 10 to 3, the average recognition speed can generally be doubled with comparable model accuracy.

## CONCLUSIONS

In this article, we propose a simple yet effective lightweight strategy for the ASR of Mandarin ATC communications, named Target-Swap Knowledge Distillation (TSKD), which swaps the logit output of the teacher and student models for the target class. TSKD consists of two components: TKD and SKD. TKD enables the student model to focus on distilling knowledge from non-target classes, while SKD mitigates the potential overconfidence of the teacher model regarding the target class.

Extensive experiments are conducted to demonstrate the effectiveness of the proposed TSKD in homogeneous and heterogeneous architectures. The experimental results indicate that either TKD or SKD contributes to the efficacy of KD, with TKD presenting a more significant benefit, validating that NCKD is the critical factor to the success of logits-based knowledge distillation. In addition, the target class is superior to the top-k predictive output for logit exchange. Moreover, the TSKD enhances the recognition accuracy of all student models by an average of 1.13%, achieving the most effective distillation results in most experiments. In particular, the optimal knowledge distillation performance of 3.98% is achieved when the ratio of TKD to SKD is 1:1 in heterogeneous architectures from

Transformer to Conformer. By restricting the beam search size from 10 to 3, the average recognition speed can generally be doubled at the cost of negligible performance loss.

In summary, with the help of TSKD, the generated lightweight ASR model balances recognition accuracy and transcription latency, allowing ATCOs and pilots sufficient time to respond immediately and effectively, thereby reducing the potential flight safety risks associated with miscommunications.

### Funding
This work was supported by the Shenzhen Science and Technology Program (No. RCBS20221008093121051), the General Higher Education Project of Guangdong Provincial Education Department (No. 2020ZDZX3085), China Postdoctoral Science Foundation (No. 2021M703371) and the Post-doctoral Foundation Project of Shenzhen Polytechnic (No. 6021330002K). The funders had no role in study design, data collection and analysis, decision to publish, or preparation of the manuscript.

### Grant Disclosures
The following grant information was disclosed by the authors:
Shenzhen Science and Technology Program: No. RCBS20221008093121051.
The General Higher Education Project of Guangdong Provincial Education Department: No. 2020ZDZX3085.
China Postdoctoral Science Foundation: No. 2021M703371.
the Post-doctoral Foundation Project of Shenzhen Polytechnic: No. 6021330002K.

### Competing Interests
The authors declare there are no competing interests.

### Author Contributions
- Jin Ren conceived and designed the experiments, performed the experiments, analyzed the data, performed the computation work, prepared figures and/or tables, authored or reviewed drafts of the article, funding acquisition, and approved the final draft.
- Shunzhi Yang conceived and designed the experiments, performed the experiments, analyzed the data, performed the computation work, authored or reviewed drafts of the article, and approved the final draft.
- Yihua Shi conceived and designed the experiments, performed the experiments, analyzed the data, authored or reviewed drafts of the article, data curation, funding acquisition, project administration, and approved the final draft.
- Jinfeng Yang analyzed the data, authored or reviewed drafts of the article, data curation, project administration, and approved the final draft.

### Data Availability
The ATCC Dataset is available at GitHub and Zenodo:

- https://github.com/Jason-6/TSKD/tree/master/data/atcc.

- Ren, Jin, Yang, Shunzhi, Shi, Yihua, & Yang, Jinfeng. (2023). A lightweight speech recognition method with target-swap knowledge distillation for Mandarin air traffic control communications. [Data set]. Zenodo. https://doi.org/10.5281/zenodo.8371514

The code is available at GitHub and Zenodo:

- https://github.com/Jason-6/TSKD

- Ren, Jin, Yang, Shunzhi, Shi, Yihua, & Yang, Jinfeng. (2023). A lightweight speech recognition method with target-swap knowledge distillation for Mandarin air traffic control communications. [Data set]. Zenodo. https://doi.org/10.5281/zenodo.8371464

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
