# Peer review of "A lightweight speech recognition method with target-swap knowledge distillation for Mandarin air traffic control communications"

_PeerJ Computer Science, doi:10.7717/peerj-cs.1650_

## Round 0.1 · original submission · Minor Revisions

This manuscript proposes an approach to improve the effectiveness of knowledge distillation by reducing the size of ASR models. However, a number of revisons are necessary. Please revise this manuscript based on the raised comments. It will be sent out for peer-review again.

**Language Note:** PeerJ staff have identified that the English language needs to be improved. When you prepare your next revision, please either (i) have a colleague who is proficient in English and familiar with the subject matter review your manuscript, or (ii) contact a professional editing service to review your manuscript. PeerJ can provide language editing services - you can contact us at copyediting@peerj.com for pricing (be sure to provide your manuscript number and title). – PeerJ Staff

Reviewer 1 ·

Basic reporting

The paper presented a well-done introduction and an excellent survey for related work, which make it clear why speech recognition for air traffic control communications requires lightweight models and low latency.The proposed simple yet effective knowledge distillation method is described clearly and illustrated well with excellent figures.Extensive experimental results show that this method can effectively reduce the low latency of student models while maintaining the recognition accuracy.

Experimental design

This paper mainly introduces a lightweight speech recognition method for air traffic control communications, which combines target-swap knowledge distillation technology to improve the generalization ability of lightweight models while maintaining high recognition performance and low latency. This paper introduces the realization process of this method in detail, including two components of knowledge distillation: TKD and SKD.

Validity of the findings

Extensive experiments were designed and conducted by authors to validate the effectiveness of the proposed method in homogeneous and heterogeneous architectures, which adopted the latest neural networks, including Transformer and Conformer. They also took the SOTA method, like DKD, as a comparison.In the comparative experiment, the SER value of the method proposed by the author is lower than that of other methods, demonstrating its superiority.Meanwhile,The optimal knowledge extraction performance in their method reached 3.98%, achieving a balance between recognition accuracy and transcription latency.

Reviewer 2 ·

Basic reporting

The authors have proposed a method that can achieve high-speed transcription while maintaining a specific recognition rate, emphasizing transcription speed, against the background that miscommunication between ATCOs and pilots causes serious accidents. This research is in the critical field of aviation accident mitigation, and its contribution to society is immeasurable.

Experimental design

Based on the constructed Mandarin ATC communications dataset and following the provided implementation details, extensive experiments were designed and conducted thoroughly to validate the effectiveness of the proposed TSKD knowledge distillation method, including comparative experiments in homogeneous and heterogeneous architectures through swapping target class or top-k predictive information, parameter sensitivity experiments, and the effect of beam search size on recognition accuracy and transcription speed.

Validity of the findings

The authors have also appropriately described the comparison with previous studies in this paper, and there is no doubt about the effectiveness of the proposed method. With the assistance of TSKD, the developed lightweight ASR model reaches a good balance between recognition accuracy and transcription speed. This enables ATCOs and pilots to react promptly and efficiently, ultimately diminishing the potential hazards to flight safety linked to communication errors.
On the other hand, I judged the paper to be conditionally accepted after minor revisions because it needs an explanation of the core terminology of the method, and the credibility of its claims needs to be better. Below are the conditions for acceptance:
(1) The difference between TCKD and NCKD cannot be distinguished because there is no detailed definition or explanation of “target” or “target class”, which is the basis of the method proposed in this paper. Please add definitions of target or target class.
(2) The correlation between the experimental results shown in the tables and the result analysis should be further refined, and their logical correlation should be more compact and direct.

Reviewer 3 ·

Basic reporting

This manuscript proposes an approach to improve the effectiveness of knowledge distillation by reducing the size of ASR models. In the author's method, elements in logits from the outputs of the teacher and student models corresponding to the target class at each time frame are swapped with each other, and the KL divergence between the probability distributions calculated from these modified versions of the logits is minimized via training. I appreciate its good survey of ATC speech recognition in Sections 1 and 2.

Experimental design

Furthermore, the proposed method was thoroughly evaluated through air traffic control (ATC) speech recognition experiments in homogeneous and heterogeneous architectures, parameter sensitivity experiments, and the beam search size’s effect on recognition accuracy and transcription speed.

Validity of the findings

The findings of this paper are correct and reliable. However, the logical relation between the experimental results and analysis should be polished to be more friendly and clear to readers. Besides, the “target class” notion should be enhanced further, which may mislead readers. Furthermore, authors are suggested to examine the grammar throughout the paper thoroughly when revising their paper.

Additional comments

no

---

## Round 0.2 · accepted · Accept

The authors have responded clearly to all comments and updated the manuscript accordingly. Thus, this paper is suitable for publication.

Reviewer 2 ·

Basic reporting

The current version is well written. All of the concerns have been well addressed.

Experimental design

The research meets the aim and scope of the journal well.

Validity of the findings

The novelty of the article is clear. All data have been provided.

Additional comments

The article is suggested to be accepted in the current version.

Reviewer 3 ·

Basic reporting

The paper can be accepted in the present form

Experimental design

The paper can be accepted in the present form

Validity of the findings

The paper can be accepted in the present form

Additional comments

The paper can be accepted in the present form